# Integrating Sequence- and Structure-Based Similarity Metrics for the Demarcation of Multiple Viral Taxonomic Levels

**DOI:** 10.3390/v17050642

**Published:** 2025-04-29

**Authors:** Igor C. dos Santos, Rebecca di Stephano de Souza, Igor Tolstoy, Liliane S. Oliveira, Arthur Gruber

**Affiliations:** 1Escola de Artes, Ciências e Humanidades, Universidade de São Paulo, São Paulo 038288-000, Brazil; igor223tgec@gmail.com; 2Instituto de Biociências, Universidade de São Paulo, São Paulo 03828-000, Brazil; rebeccadistephano@usp.br; 3Argentys Informatics, LLC, 12 South Summit Avenue Suite 200, Gaithersburg, MD 20877, USA; itolstoy@gmail.com; 4Department of Computer Science, Federal University of Technology of Paraná (UTFPR), Alberto Carazzai Avenue, 1640, Cornélio Procópio 86300-000, Brazil; liliane.sntn@gmail.com; 5Department of Parasitology, Instituto de Ciências Biomédicas, Universidade de São Paulo, São Paulo 05508-000, Brazil; 6European Virus Bioinformatics Center, Leutragraben 1, 07743 Jena, Germany

**Keywords:** viral classification, viral taxonomy, taxa demarcation, sequence similarity, protein structure similarity

## Abstract

Viruses exhibit significantly greater diversity than cellular organisms, posing a complex challenge to their taxonomic classification. While primary sequences may diverge considerably, protein functional domains can maintain conserved 3D structures throughout evolution. Consequently, structural homology of viral proteins can reveal deep taxonomic relationships, overcoming limitations inherent in sequence-based methods. In this work, we introduce MPACT (Multimetric Pairwise Comparison Tool), an integrated tool that utilizes both sequence- and structure-based metrics. The program incorporates five metrics: sequence identity, similarity, maximum likelihood distance, TM-score, and 3Di-character similarity. MPACT generates heatmaps and distance trees to visualize viral relationships across multiple levels, enabling users to substantiate viral taxa demarcation. Taxa delineation can be achieved by specifying appropriate score cutoffs for each metric, facilitating the definition of viral groups, and storing their corresponding sequence data. By analyzing diverse viral datasets spanning various levels of divergence, we demonstrate MPACT’s capability to reveal viral relationships, even among distantly related taxa. This tool provides a comprehensive approach to assist viral classification, exceeding the current methods by integrating multiple metrics and uncovering deeper evolutionary connections.

## 1. Introduction

Viral diversity far exceeds that of cellular organisms. Viruses exhibit a wide range of genomic compositions, consisting of either DNA or RNA, which may be single- or double-stranded, and can exist in either sense or antisense orientations. These genomes can be organized into single or multiple segments and vary significantly in size, encapsulated within capsids of diverse morphologies [1]. Some viruses may lack capsids entirely, persisting as extrachromosomal elements within host cells [2,3]. While eukaryotic organisms trace their lineage to a last universal common ancestor (LUCA), distinct from bacteria and archaea, viruses exhibit profound evolutionary divergence and may have emerged multiple times throughout evolutionary history prior to the origin of LUCA [4,5]. Unlike cellular life forms, which possess universally conserved evolutionary markers such as 16S or 18S rRNA genes, viruses lack common genetic markers, complicating phylogenetic reconstruction [6,7].

The classical classification system proposed by David Baltimore over 50 years ago [8] was based on the transmission pathways of viral genome information and initially comprised six groups, later expanded to seven. Despite significant advancements in virology, this classification remains a foundational concept for understanding information transmission pathways in biological systems [9]. Nevertheless, taxonomic assignments frequently rely on variable and inconsistent criteria across diverse viral groups, and continue to be a matter of ongoing debate [1]. In contrast to the well-structured taxonomy of cellular organisms, a formalized binomial Latin nomenclature for viruses was only proposed in 2020 [10], with implementation recommendations still in progress [11].

Traditional viral classification, established in the mid-20th century, included only five taxonomic levels: species, genus, subfamily, family, and order. This model primarily aimed to group closely related viruses. Recently, the International Committee on Taxonomy of Viruses (ICTV) introduced a hierarchical classification system with 15 taxonomic ranks to accommodate the vast genetic diversity of the virosphere, thereby enabling classifications that reflect basal evolutionary relationships among distantly related viruses [12]. The highest rank, ‘realm’, is analogous to the domain level in cellular life taxonomy and reflects the complex interplay between viral and cellular taxonomies. Realms do not share a universal common ancestor but group viruses with shared genetic traits. Within this framework, members of each realm share sets of ancestral orthologous genes, typically associated with replication or virion formation. This perspective acknowledges that viral classification may extend beyond taxonomy to include alternative classification schemes based on clinical or epidemiological properties.

A panel of virology taxonomy experts proposed four guiding principles for viral classification [13]. The first principle mandates that all the recognized taxa must be monophyletic. The second principle suggests that phenotypic and ecological traits may be informative but should not override phylogenetic reconstruction. The third principle asserts that taxonomic classification is just one approach to categorizing viruses, and alternative classifications based on infectivity and virulence may be valuable for disease prevention and treatment, despite often forming polyphyletic groups that do not reflect evolutionary relationships. Lastly, the fourth principle emphasizes the importance of quality control, particularly in metagenomic data, for assigning any viral taxonomic classification.

With the rapid expansion of viral genome sequencing, particularly from metagenomic samples, the development of computational tools for viral sequence classification has become essential. A diverse array of software applications employing various methodological approaches is currently available [14]. One classification approach relies on genetic content conservation and genome organization in gene or protein profiles [15]. Notable platforms utilizing this approach include GRAViTy [16,17] and vConTACT v.2.0 [18]. vConTACT v.2.0 applies network-based whole-genome gene-sharing profiles to perform hierarchical clustering with integrated confidence scores, enabling automated taxonomic classification at the genus and basal taxonomic levels. GRAViTy considers gene content, orientation, and protein-coding signatures, calculating composite generalized Jaccard distances (CGJ) to group viruses into taxonomic categories, spanning from families to inter-family relationships [16,17]. Another category of tools employs phylogenetic reconstruction to establish monophyletic clades for classification. This widely accepted method produces consistent taxonomies, provided that the sequences retain phylogenetic signal. However, due to the high evolutionary rates and substantial divergence among viral families, this approach, as implemented in platforms such as VICTOR [19], is less effective at resolving higher taxonomic ranks, such as families and orders.

Pairwise sequence distance is among the metrics favored by the ICTV for establishing a universal viral classification method. A key challenge in this approach is defining optimal distance thresholds for each taxonomic level, considering the varying genetic divergence across taxa. The PASC (PAirwise Sequence Comparison) tool [20], available through the NCBI (National Center of Biotechnology Information), compares query sequences against a viral sequence database using BLAST or the Needleman–Wunsch algorithm, generating frequency distributions of genetic identity scores to facilitate taxonomic delineation. The DEmARC (DivErsity pArtitioning by hieRarchical Clustering) tool [21] utilizes multiple sequence alignments and evolutionary distance calculations based on probabilistic models, verifying monophyly and optimizing distance thresholds to classify viruses across all the taxonomic levels within a monophyletic group. This tool is particularly suited for nucleotide sequence analysis and is optimized for studying protein sequence domains. The Sequence Demarcation Tool (SDT) performs pairwise alignments using a Needleman–Wunsch (NW) approach that disregards indel-containing positions [22]. It integrates with the Neighbor program of the PHYLIP package to construct a rooted Neighbor-joining phylogenetic tree, organizing sequences based on inferred evolutionary relationships. SDT generates heatmaps to visualize pairwise identity distributions, facilitating intuitive data interpretation. While designed for nucleotide sequence analysis, SDT can also be used with amino acid sequences, although it is then limited to calculating identity percentage, rather than amino acid sequence similarity.

For the detection and classification of novel viruses, pairwise similarity searches, such as BLAST are the most commonly used method, but distant relationships cannot be detected by any pairwise comparison method [23]. Profile-based methods like PSI-BLAST with position-specific scoring matrices (PSSMs) and profile hidden Markov models (HMMs), are more sensitive [6,7,24,25] and can detect distantly related viruses. However, with the increasing influx of metagenomic viral sequences, a vast array of highly divergent viruses is being discovered, often precluding classification restricted to sequence similarity. Protein structure is much more conserved along evolution than primary sequences, allowing to reveal relationships across evolutionary remote organisms [26]. Novel approaches leveraging protein domain structure have gained attention. Advances in artificial intelligence and protein structure prediction, exemplified by AlphaFold [27,28] and ESMFold [29], have led to the development of extensive 3D protein structure databases, such as AlphaFold DB [28] and the ESM Metagenomic Atlas. Efficient structural similarity search tools, including DALI [30] and TM-align [31], have emerged, with the recently developed Foldseek [32] significantly reducing computational requirements while maintaining high sensitivity [33].

The delineation of viral taxonomic levels remains one of the major challenges in virus taxonomy and classification. Currently, there are no simple and versatile tools that integrate multiple metrics for the graphical visualization of viral taxonomic distances and classification. The development of a simple and integrated tool that incorporates multiple metrics and provides graphical output in the form of heatmaps would be highly beneficial for the rapid visualization of viral groupings. The integration of various distance metrics—such as nucleotide and amino acid similarity, maximum likelihood (based on evolutionary models), and three-dimensional structural distance—within a single platform is unprecedented and could constitute a significant contribution to the virology research community by adding quantitative and qualitative information to assist viral classification.

In this work, we aimed to develop an integrated tool for the analysis and graphical visualization of biological sequence distance data and 3D protein structures from multiple organisms to support the delineation of viral taxonomic levels. We introduce MPACT, the Multimetric PAirwise Comparison Tool. Applications of this program across diverse viral groups, encompassing varying evolutionary distances, are demonstrated, highlighting its capacity to generate multiple graphical outputs and partition sequence data based on objective parameters.

## 2. Materials and Methods

### 2.1. Data Sources

#### 2.1.1. RNA Virus Families of the *Orthornavirae* Kingdom (*Riboviria* Realm—RNA Viruses)

We used 107 sequences of the RNA-directed RNA polymerase representing different families of the *Orthornavirae* kingdom (*Riboviria* realm—RNA viruses and retroviruses) [34], including *Totiviridae* (phylum *Duplornaviricota*), *Amalgaviridae* (phylum *Pisuviricota*) [35], *Partitiviridae* (phylum *Pisuviricota*) [36], and four families of the phylum *Lenarviricota*: *Mitoviridae*, *Botourmiaviridae*, *Narnaviridae*, and *Leviviridae* (Appendix A). Two additional datasets comprising 66 sequences of the ORF1 protein (Appendix A) and 67 sequences of the RDRP (Appendix A) of *Amalgaviridae* viruses were also used. All the protein sequences were obtained from public sources. Since *Amalgaviridae* viruses express fusion proteins [37,38], including the ORF1 protein and the RDRP, these sequences were manually separated for the respective datasets.

#### 2.1.2. *Microviridae* Family (*Monodnaviria* Realm—ssDNA Viruses, *Sangervirae* Kingdom)

We used a dataset composed of 119 sequences of the major capsid protein (VP1) from representatives of the different groups/subfamilies of the *Microviridae* family reported and proposed in the literature, which are not officially recognized by the current ICTV classification. The dataset comprises sequences of *Alpavirinae* [39], *Gokushovirinae* [40,41], *Pichovirinae* [39], *Sukshmavirinae* [42], Group D [43], Parabacteroidetes prophage [44], *Aravirinae* [44], *Stokavirinae* [44], *Liberivirinae* [45], *Amoyvirinae* [46], *Bullavirinae* [47] *Pequeñovirus* [48], CGM group [49], *Tainaviridae* [50], *Occultatumvirinae* [50], *Reekeekeevirinae* [51], and *Roodoodoovirinae* [51]. The accession codes and sources of the sequences are listed in Appendix A.

#### 2.1.3. *Orthobunyavirus* Genus (*Negaraviricota* Kingdom—Negative-Strand ssRNA Viruses, *Riboviria* Realm)

We used *Orthobunyavirus* datasets containing 55 nucleotide sequences of the large (L) segment and their respective translated amino acid sequences. These datasets comprise representative isolates of 14 distinct serogroups of the genus *Orthobunyavirus* [52] (*Peribunyaviridae* family). NCBI accession codes of the sequences are listed in Appendix A.

### 2.2. Multiple Sequence Alignment and Phylogenetic Analysis

Multiple sequence alignments (MSAs) were generated using MAFFT v7.505 [53]. Phylogenetic reconstruction was performed using IQ-TREE v2.2 [54], with the ModelFinder [55] program used to determine the model that minimizes the Bayesian Information Criterion (BIC) score. Node support values were determined using 1000 pseudoreplicates with the ultrafast bootstrap approximation (UFBoot) method [56].

### 2.3. Three-Dimensional Protein Structure Prediction

The protein sequences of the datasets were used to predict their respective 3D structures. The AlphaFold2 platform [27,28] was employed, specifically using the ColabFold v1.5.5 server: AlphaFold2 [57]. From the five 3D structures generated by AlphaFold, the top-ranked structure based on the predicted local distance difference test (pLDDT) value [58] was selected for downstream analyses. PDB structure files were converted to 3Di-character files using Foldseek [32].

### 2.4. Implementation of MPACT Program

MPACT, the Multimetric PAirwise Comparison Tool is an integrated toolbox for pairwise all-against-all comparison of primary nucleotide or amino acid sequences, and 3D protein structures. For nucleotide sequences, the program utilizes identity percentage and maximum likelihood (ML) distance as metrics. In the case of proteins, MPACT determines the identity percentage, similarity percentage, and maximum likelihood distance of amino acid sequences, and also obtains structural similarity (TM-scores) and 3Di-character similarity of 3D structures. For each metric, the program performs data clustering and generates heatmaps, frequency distribution plots, and dendrograms. Also, MPACT can partition sequence datasets according to user-defined criteria for each metric. MPACT is written in Python 3 (https://www.python.org/, accessed on 1 March 2025) and utilizes the following third-party programs: Needle (from the EMBOSS package version 6.6.0 [59]), MAFFT [53], IQ-TREE 2 [54], Foldseek [32], and TM-align [31]. The program executable, usage manual, tutorial, and a Docker container image for easy execution are available on GitHub (https://github.com/gruberlab/mpact, accessed on 1 March 2025).

## 3. Results

### 3.1. Workflow of the Program

The MPACT program’s workflow is depicted in Figure 1. The program utilizes two types of input data: protein or nucleotide sequences in the FASTA format, and 3D protein structure files in the PDB format. The biological sequences are submitted to all-against-all pairwise alignments using the Needleman–Wunsch global alignment algorithm implemented in the Needle program (EMBOSS package). MPACT extracts global identity and similarity percentage values from these alignments and stores the results as matrices. The input sequences are also submitted to a multiple sequence alignment (MSA) with MAFFT. The resulting MSA is then used for ML phylogenetic analysis using IQ-TREE 2, and the resulting ML phylogenetic tree and distance matrix are stored. Three-dimensional protein structures are submitted to all-against-all pairwise structural comparisons using TM-align. All the resulting pairwise scores, normalized by the average length of the compared structures, are used to create a TM-score matrix. The PDB files are also converted to 3Di-character sequences by the Foldseek program, and these sequences are aligned using Needle with a 3Di substitution matrix [32]. Data clustering is performed on all the matrices using the clustermap function from the Seaborn 0.12.2 data visualization library. The resulting clustering trees (Newick format) and heatmap images (JPG and SVG) are stored. Additionally, MPACT converts the matrix data into range-tabulated values and creates frequency distribution plots with Seaborn’s lineplot function. The final section of the workflow is devoted to data partitioning. In the first step, all the matrices are converted to distance matrices. The data are then submitted to Neighbor-joining (NJ) hierarchical clustering using Biopython’s Bio.Phylo package. The generated NJ tree is rooted at the midpoint, and its nodes are ordered in increasing order of branch length. This order is then used for data partitioning to generate groups of sequences selected within a range of user-defined upper and lower values for each chosen metric. Finally, MPACT stores the resulting tree, heatmap, and sequence files.

### 3.2. Using MPACT on Viral Protein Sequence and 3D Structure Datasets

#### 3.2.1. Application on RNA Viral Families of the *Orthornavirae* Kingdom

To assess the ability of the MPACT program to unravel relationships across different viral families, we chose first to analyze several RNA virus families belonging to some phyla of the kingdom *Orthornavirae* (*Riboviria* realm—RNA viruses and retroviruses). These included eukaryotic double-stranded RNA (dsRNA) viruses from the families *Totiviridae* (phylum *Duplornaviricota*), *Amalgaviridae* (phylum *Pisuviricota*), and *Partitiviridae* (phylum *Pisuviricota*). Both *Totiviridae* and *Amalgaviridae* have monopartite genomes, whereas *Partitiviridae*, commonly found in plants and fungi, typically possess bipartite genomes with two RNA segments. *Totiviridae* primarily infect fungi, protozoa, and invertebrates, and have genomes ranging from 4.6 to 7.0 kb that encode two open reading frames (ORFs): one for the capsid protein (CP) and another for the RNA-dependent RNA polymerase (RDRP). In *Partitiviridae*, the CP and RDRP are encoded on separate genome segments. While *Totiviridae* and *Partitiviridae* form non-enveloped icosahedral capsids approximately 30–40 nm in diameter, *Amalgaviridae* are believed to exist as non-encapsidated RNA–protein complexes within host cells. *Amalgaviridae* infect plants, fungi, and invertebrates, and their 3.4–3.5 kb genomes encode an ORF1 protein of unknown function and an RDRP. In addition, we included positive-strand single-stranded RNA [(+)ssRNA] eukaryotic viruses of the phylum *Lenarviricota*, comprising the families *Mitoviridae*, *Narnaviridae*, and *Botourmiaviridae*, as well as two representatives from the *Leviridae* family, which are (+)ssRNA viruses that infect prokaryotes. The three eukaryotic viral families have genomes ranging from 2 to 3.6 kb and each contains a single open reading frame (ORF) that encodes for the RNA-dependent RNA polymerase (RDRP). These viruses infect plants and fungi, and in the cases of *Mitoviridae* and *Narnaviridae*, they have also been found in invertebrates. Notably, no virions are produced by these viruses. Finally, we used representatives of the *Leviviridae* family as an outgroup. These viruses have an icosahedral capsid of 28 to 30 nm in diameter, contain a (+)ssRNA genome, and primarily infect prokaryotes, mainly bacteria.

To guide our analyses as a golden standard, we performed a phylogenetic reconstruction (Figure 2) of these viral families using RDRP sequences. All the families showed monophyly, with the dsRNAviruses presenting a major clade composed of the diverse group of monopartite Amalgaviridae and a sister clade of the bipartite Partitiviridae. A more basal clade comprises monopartite viruses of the Totitiviridae family. The (+)ssRNA viruses constitute a major sister clade containing a large subclade with three families of eukaryotic viruses (*Mitoviridae*, *Narnaviridae*, and *Botourmiaviridae*), and a more external clade comprising the prokaryotic Leviviridae phages. These evolutionary relations are in agreement with literature reports [3,35,60,61]. This protein dataset was processed by the MPACT program using the five metrics. Figure 3 shows a typical output generated by MPACT, displaying a heatmap graph of ML distance values, with an upper UPGMA dendrogram. Most of the relationships observed in the phylogenetic analysis (Figure 2) are congruent with the clusters obtained for the ML distance heatmap (Figure 3). MPACT also generates frequency distribution plots (Appendix A) allowing for the comparison of the five metrics, an output that can be used to support data partitioning criteria.

To investigate how different metrics reflect the diversity of viral taxa and protein markers, we restricted our analysis to two proteins from viruses of the *Amalgaviridae* family. This family includes the genus *Amalgavirus*, a group of important plant pathogens, and several Amalga-like viruses found in diverse other hosts, primarily fungi and some insects. The monopartite genome of this family, similar to that of the related sister family *Totiviridae*, comprises two open reading frames (ORFs). A key difference between the *Amalgaviridae* and *Totiviridae* families lies in the first ORF, which encodes a capsid protein in *Totiviridae* but a protein of unknown function in *Amalgaviridae*. The second ORF codes for the RDRP in both families.

The results obtained using MPACT clearly show that amino acid identity percentage is by far the least conserved metric across amalgaviruses for both proteins, displaying heatmaps with the coolest colors (Figure 4(A1),(B2)). A higher level of conservation is observed for amino acid similarity (Figure 4(B1),(B2)), reflecting evolutionary constraints that conserve residues sharing physicochemical properties. Such residues may play equivalent roles in a functional domain or contribute to protein structure stabilization. Much higher conservation is observed for ML distance in both ORF1 protein (Figure 4(C1)) and RDRP (Figure 4(C2)). To understand these results, it is important to consider how the ML distance matrix is calculated. MPACT uses MAFFT to generate an MSA of the viral sequences and then executes IQ-TREE 2 for phylogenetic reconstruction. IQ-TREE 2 employs an ML model to estimate evolutionary relationships, determining the tree topology and branch lengths that maximize the likelihood of the observed data given the model. These branch lengths are then used to calculate pairwise evolutionary distances, which comprise the ML distance matrix. Therefore, ML distance closely reflects phylogenetic analysis. This represents a substantial improvement over simple identity and similarity percentages, which are calculated primarily from pairwise alignments and lack a basis in an evolutionary model.

The structural data comparisons across *Amalgaviridae* viruses show discrepant results between the ORF1 protein (Figure 4(D1)) and RDRP (Figure 4(D2)) for the TM-score metric. The ORF1 protein presents an overall low structural similarity characterized by a heatmap with cool colors, whereas RDRP shows much higher TM-scores depicted by warmer colors. This result suggests that the ORF1 protein’s function does not depend on a very rigid and stable structure. On the other hand, RDRP demonstrates high structural conservation, which is compatible with its highly specialized enzymatic function, the replication of the genetic material of RNA viruses. For both proteins, 3Di-character sequence similarity heatmaps (Figure 4(E1),(E2)) are relatively similar to the ML distance (Figure 4(C1),(C2)) heatmaps. An intriguing observation arises from the unexpected discrepancy between TM-score (Figure 4(D1)) and 3Di similarity heatmaps (Figure 4(E1)) for the ORF1 protein. Given that 3Di-characters represent a discretization of protein structures, one would anticipate higher all-against-all conservation in 3Di-character similarity heatmaps compared to amino acid similarity, yet lower than that observed for TM-scores. In fact, the TM-score heatmap of RDRP (Figure 4(D2)) exhibits higher conservation than the corresponding 3Di-character similarity (Figure 4(E2)), which in turn is higher than amino acid similarity (Figure 4(B2)). Conversely, for the ORF1 protein, 3Di-character similarity (Figure 4(E1)) displays significantly greater cross-conservation than both, amino acid similarity (Figure 4(B1)) and TM-score (Figure 4(D1)). This discrepancy can be attributed to the distinct alignment strategies employed in the analyses by TM-align and 3Di-character similarity. While TM-align focuses on globally aligning relatively rigid structures, 3Di-character analysis emphasizes the geometric relationships between adjacent residues [32]. Consequently, 3Di-character alignment tends to prioritize shorter structural segments and the identification of local interactions between spatially proximate amino acids, potentially overlooking the overall 3D structural alignment.

To better understand this result, we performed 3D protein structure predictions of the ORF1 proteins using AlphaFold2 and the results confirmed a high abundance of alpha-helical regions (Appendix A). The pLDDT (predicted local distance difference test) values for the five top-ranked 3D structures ranged from 51.0 to 78.2, while the pTM (predicted template modeling) scores ranged from 0.342 to 0.560. For comparison, the structure predictions of the RDRPs from the same organisms yielded pLDDT values ranging from 59.0 to 86.5 and pTM values ranging from 0.615 to 0.865. These results corroborate that the ORF1 protein presents a more relaxed/flexible 3D structure than the RDRP. The precise function of the ORF1 protein remains unknown and, despite numerous efforts, viral particles have not been observed in amalgaviruses, unlike totiviruses, which exhibit icosahedral capsids [62]. The ORF1 protein shows no similarity to capsid proteins and some evidence points out a possible interaction with the viral RNA genome and a potential protective role [35]. Also, immunogold electron microscopy revealed that amorphous bodies in the cytoplasm of blueberry cells and these aggregates could be involved in viral genome protection [62]. In conclusion, the ORF1 protein may lack a stable, well-defined 3D structure and could exhibit high flexibility and/or a tendency to aggregate. This characteristic, coupled with the abundance of alpha-helical domains, could explain the discrepant results obtained by MPACT using TM-align structural similarity (Figure 4(D1)) and 3Di-character similarity (Figure 4(E1)).

#### 3.2.2. Application on Bacteriophages of the *Microviridae* Family

We extended the validation of the MPACT program to *Microviridae*, a family of phages that infect a wide range of bacterial hosts across various environments. This viral family comprises a growing number of groups, whose taxonomic classification remains incomplete [63]. To establish a golden standard overview of the family’s evolutionary relationships, we utilized VP1 (major capsid protein) sequences from representatives of the numerous subfamilies/groups reported in the literature and accessible through public repositories. The ML phylogenetic reconstruction (Figure 5) demonstrates that these groups are monophyletic, with some exhibiting relatively long branches, indicative of high divergence and evolutionary rate. It is noteworthy that phages classified as members of the *Pichovirinae* [39] subfamily and the Parabacteroidetes group [44] are closely related. Likewise, phages belonging to the so-called CGM group [49] are part of the proposed *Occultatumvirinae* subfamily, which constitutes a sister clade to the *Tainavirinae* subfamily, both of them hosted by Alphaproteobacteria [50]. Our findings are in agreement with the original description of the *Reekeekeevirinae* subfamily [51], which constitutes a basal clade to *Liberivirinae* [45] and *Amoyvirinae* [46], and with the phylogenetic location of the *Roodoodoovirinae* subfamily [51], forming a more basal clade to *Bullavirinae* and *Pequeñovirus*. Insertion loops from three VP1 subunits constitute mushroom-like protrusions that are observed in the viral capsids of some *Microviridae* phages [64]. Our 3D protein structure predictions (Appendix A) confirm the existence of insertion loops in the capsid protein (VP1), as originally reported for *Alpavirinae*, *Gokushovirinae* and *Pichovirinae* [39], *Stokavirinae* and *Aravirinae* [44], and *Reekeekeevirinae* and *Roodoodoovirinae* [51]. We also observed these loops in members of *Sukshmavirinae*, Group D, Parabacteroidetes prophages, *Occultatumvirinae*, and *Tainavirinae*, and this is the first report of protrusion loops in members of these *Microviridae* subfamilies/groups. Our phylogenetic analysis (Figure 5) reveals an interesting correlation between the main clades and the presence or absence of insertion loops that form mushroom-like protrusions. Beginning from the most basal clades, which correspond in ascending order to *Reekeekeevirinae*, *Amoyvirinae*, and *Liberivirinae*, no mushroom-like structures are observed. Conversely, the sister clade to these subfamilies, comprising *Occultatumvirinae*/CGM group and *Tainavirinae*, exhibits the insertion loop on the VP1 structure. Continuing in ascending order, we observe a clade encompassing *Roodoodoovirinae*, *Pequeñovirus*, and *Bullavirinae*, which lack mushroom-like structures. Finally, all the remaining subfamilies/groups within the upper sister clade possess insertion loops on VP1. These results suggest that mushroom-like structures may have originated independently on multiple occasions throughout the evolutionary history of *Microviridae* phages, but the functional role of these structures remains unknown.

Heatmaps generated by MPACT using five distinct metrics reveal that ML distance (Figure 6C) more effectively highlights the similarities between members of each group compared to amino acid similarity (Figure 6B) or identity (Figure 6A) percentages, consistent with the results observed for the ORF1 protein and RDRP of viruses within the *Orthornavirae* kingdom (Figure 4). As previously mentioned, ML distance closely reflects the phylogenetic analysis and evolutionary history of the viral groups. The TM-align scores (Figure 6D) and 3Di-character similarity (Figure 6E) clearly display the overall strongest relationships across the different taxa, even for those that are more distantly related. This is expected, since both metrics are based on the conservation of 3D structures, which is more conserved than primary sequence identity or similarity percentages [26], and VP1 presents structural constraints due to its role in capsid formation.

### 3.3. Viral Group Demarcation

One of the primary applications of comparing viral nucleotide or protein sequences across different taxa is to establish criteria for taxonomic delineation. An ideal demarcation criterion should generate reliable viral groups with few or no misplaced taxa within the respective clusters. Phylogenetic analysis, grounded on evolutionary models, provides a robust framework for evaluating and guiding manual assignments. Since MPACT uses IQ-TREE 2 to perform phylogenetic reconstruction, we obtained ML phylogenetic trees for our datasets of *Orthornavirae* RNA viruses (Figure 2) and *Microviridae* phages (Figure 5). Both viral protein datasets yielded well-resolved trees, characterized by monophyletic groups and strong node support values.

As MPACT executes several analyses using various programs on both sequence and 3D structure data, it enables the selection of the most informative and discriminative metrics to assist taxa demarcation. Starting from the distance matrices obtained for the various metrics, it is necessary to define value range limits for the clustering task. For instance, given a similarity percentage matrix, we need to delineate upper and lower percentage values as inclusion and exclusion criteria for the groups. Since the optimal values are not available a priori, we performed inter- and intragroup comparisons. Members of the viral taxa were defined by the maximum likelihood (ML) phylogenetic reconstructions and data reported in the literature, resulting in bona fide subsets. For each subset, sequences were compared in an all-against-all fashion using the different metrics to determine the intragroup value range for each metric. Sequences from each subset were also compared to the remaining sequences of the whole dataset to determine the intergroup value range. This approach was employed for the groups of RNA viruses (Appendix A) and *Microviridae* phages (Appendix A).

Across the different families of *Orthornavirae* viruses (Appendix A), all the metrics exhibited a wide range of intragroup values. For example, within the *Amalgaviridae* family, amino acid sequence identity and similarity of the RDRP varied substantially, ranging from 10.4% to 71.7% and 17.0% to 81.9%, respectively. Similar variability was observed for ML distance, TM-score, and 3Di-character sequence similarity. To investigate whether a broad spectrum of intragroup diversity is a common feature across different viral groups, we extended the study to estimate the range of intragroup diversity values within *Microviridae* (Appendix A). Like RNA viruses, the different subfamilies/groups of *Microviridae* also revealed a wide range of intragroup variability. For instance, the *Gokushovirinae* subfamily exhibited, for the VP1 protein, amino acid sequence identity and similarity values ranging from 32.6% to 88.8% and 48.9 to 93.1%, respectively. A comparison of the overall values observed for the five metrics in *Orthornavirae* (Appendix A) and *Microviridae* (Appendix A) revealed considerably higher intragroup divergence in the former dataset. The *Orthornavirae* dataset includes dsRNA viruses representatives from the phyla *Duplornaviricota* and *Pisuviricota*, as well as (+)ssRNA viruses belonging to the phylum *Lenarviricota*. In contrast, the *Microviridae* dataset is limited to (+)ssDNA viruses from different subfamilies/groups within a single family. This substantial difference in the taxonomic breadth represented by each dataset may explain the greater divergence observed within the *Orthornavirae* dataset compared to the *Microviridae* dataset. In fact, the mean pairwise similarity percentage of *Orthornavirae* viruses is 22.1% ± 10.1 (Appendix A), whereas this value for *Microviridae* phages is 31.8% ± 11.9 (Appendix A).

Although the *Microviridae* subfamilies show lower intragroup divergence than the different families of the *Orthornavirae* dataset, a considerable range of intragroup diversity is still observed. This feature is likely a consequence of the very high mutation rates of viruses [65,66,67,68,69], leading to high divergence rates even within relatively narrow taxonomic groups. The wide range of intragroup diversity observed across all the evaluated metrics in both the *Orthornavirae* (Appendix A) and *Microviridae* (Appendix A) hinders the clear demarcation of viral taxa. For every metric, regardless of the viral group considered, the range of intragroup values overlaps with the corresponding intergroup value range. For instance, *Alpavirinae* shows, for the VP1 protein, intragroup amino acid identity and similarity values ranging from 14.1% to 65.4% and 23.4% to 76.6%, respectively (Appendix A). The corresponding intergroup ranges are 2.1% to 29.3% and 2.9% to 43.4%, respectively. This indicates that the *Alpavirinae* subfamily includes members whose sequences are more similar to viruses belonging to other subfamilies than they are to members within their own subfamily. This pattern is also observed when analyzing the corresponding value ranges of ML distance, TM-scores, and 3Di-character sequence similarity (Appendix A). Similarly, overlaps between intragroup and intergroup value ranges are seen for all the families of *Orthornavirae* (Appendix A and Appendix A). These results indicate that high intragroup diversity prevents any metric from discriminating between the various viral groups within the analyzed datasets.

Based on these results, we decided to analyze a taxonomically narrower dataset, focusing on representative viruses from different antigenic groups/serogroups within the genus *Orthobunyavirus* (Appendix A). As anticipated, the mean pairwise amino acid similarity percentage of the RDRP within *Orthobunyavirus* was 71.85% ± 6.9 (Appendix A), significantly higher than the values observed for the RDRP of *Orthornavirae* (22.1% ± 10.1—Appendix A) and VP1 of *Microviridae* (31.8% ± 11.9—Appendix A). Similarly to the approach adopted for *Orthornavirae* and *Microviridae*, we used a dataset of bona fide representatives from 14 of the 18 antigenic groups/serogroups reported in the literature [70,71,72] for phylogenetic reconstruction. Since the members of the genus *Orthobunyavirus* are closely related, we used nucleotide sequences of the L segment for this analysis. The resulting phylogenetic tree (Figure 7) showed monophyly across the different groups and high node support values for most clades. It is noteworthy that some clades display short branch lengths, as observed for California, Gamboa, and Bunyamwera, while other groups show very long branches, such as Nyando, Anopheles A, Simbu, and Maputta. This result can be attributed to a variety of factors, acting either independently or in concert: (1) significantly different evolutionary rates among serogroups, (2) low taxa sampling, and (3) variations in host diversity and population dispersal. Given the relatively close relationship among these viruses, we performed MPACT analyses on both, their nucleotide sequence of the L segment and the corresponding amino acid sequences of the coded RDRP. Unlike what has been observed for *Orthornavirae* and *Microviridae*, we obtained *Orthobunyvirus* clear separation of intragroup and intergroup value ranges for the different analyzed metrics (Appendix A and Figure 8). These findings suggest that for *Orthobunyavirus*, arbitrary thresholds may serve as effective parameters for determining the inclusion or exclusion of viral members within distinct serogroup-related clusters. It is conceivable that similar effective demarcation could be achieved for other viral taxa, provided their members exhibit relatively high similarity (e.g., amino acid sequence similarity above 70%).

Collectively, our findings from the *Orthornavirae*, *Microviridae*, and *Orthobunyavirus* datasets demonstrate that viral group demarcation using fixed upper and lower value limits for any single metric can be misleading. This is due to the potential for greater variation within a viral group than between groups, a phenomenon particularly evident in higher taxonomic ranks, as observed with *Orthornavirae*. Conversely, for closely related viral groups, such as members of different serogroups of the genus *Orthobunyavirus*, clear delineation is both possible and effective, validating the ICTV’s species demarcation methodology adopted for many viruses. While single metric value limits are unsuitable for higher taxonomic ranks, MPACT’s heatmaps and dendrograms, which closely resemble phylogenetic reconstructions (e.g., Figure 3), offer multiple lines of evidence for delineation, especially when incorporating 3D structure-based comparisons.

### 3.4. Data Partitioning

To explore the potential of using specific metric values to include or exclude sequences within viral groups, we used MPACT to partition the nucleotide sequences of the L segment of the genus *Orthobunyavirus*. Based on the results obtained for intragroup and intergroup comparisons using the various metrics (Appendix A), we tested different value ranges for data partitioning and determined the resulting number of clusters (Appendix A). Considering that the original *Orthobunyavirus* dataset comprised representatives of 14 serogroups, we chose partitioning value ranges that produced a cluster number as close as possible to 14. Specifically, we chose 65–100% nucleotide identity percentage (18 clusters—Appendix A) and 0–1.5 of nucleotide ML distance (13 clusters—Appendix A). Using nucleotide ML distance, Cluster 1 contains all representatives of the California serogroup and a unique sequence from the Bwamba serogroup (Appendix A). This clustering is congruent with the phylogenetic analysis of the genus *Orthobunyavirus* (Figure 7), as the Bwamba sequence forms a basal branch to the California clade. Cluster 2 is composed solely of Gamboa sequences, similar to Clusters 9 and 13, which contain representatives of the Tete and Maputta serogroups, respectively, in agreement with the monophyly observed in the phylogenetic tree. Serogroups presenting long branches (Figure 7), as observed for the Nyando (Clusters 3 and 4), Anopheles (Clusters 6 and 7), and Simbu (Clusters 10, 11, and 12) serogroups, were separated into distinct clusters (Appendix A). Conversely, serogroups forming sister clades with short distances between them were included in the same clusters. This result was observed for Bunyamwera and Wyeomyia (Cluster 5), as well as for the Guama, Capim, Group C, and Patois (Cluster 8) serogroups.

In the case of nucleotide pairwise identity percentage, the clustering results were similar, but more stringent, leading to the separation of some clades into additional clusters (Appendix A). We also tested data partitioning using amino acid sequences of RDRP with three different metrics to assess whether this type of data would yield clusters more congruent with the serogroup clades obtained in the phylogenetic tree (Figure 7). Our results (Appendix A) showed no improvement in the partitioning process compared to the results obtained with the corresponding nucleotide sequences (Appendix A). We conclude that the clustering process itself is functioning properly, but it is not possible to obtain clusters that perfectly match the clades observed in the phylogenetic tree, regardless of the sequence type (nucleotide or amino acid), or metric chosen. The primary reason for this discrepancy is that the clustering process is limited by a single range of metric values applied to the entire dataset. However, as already commented, the evolution rate varies significantly across different clades, resulting in a wide diversity of branch lengths. Consequently, regardless of the metric, no single value range can precisely define the clades that correspond to the different serogroups within the genus *Orthobunyavirus*. Within certain ranges, clades encompass all representatives of a serogroup, while other clades may contain a mixture of closely related serogroups. Finally, some serogroups with more distantly related members may be divided into multiple clades, as was the case of the Nyando serogroup. To conclude, although taxa demarcation and data partitioning are feasible with MPACT or comparable tools, users must be keenly aware of the inherent limitations of this methodology.

### 3.5. Comparison of MPACT and Other Similar Tools: SDT and Dali

Two different publicly available tools are closely related to MPACT: the Sequence Demarcation Tool (SDT) [22] and Dali [30]. SDT was originally developed for all-against-all pairwise alignments of nucleotide sequences but it can also be used for pairwise alignments of amino acid sequences. The program generates color-coded heatmaps according to identity levels and also frequency distribution plots. Furthermore, SDT allows data partitioning through user-defined lower and upper identity percentage values, enabling easy sequence demarcation. We used a dataset composed of 55 nucleotide sequences of the large (L) segment of viruses of the *Orthobunyavirus* genus to compare MPACT and SDT. Both programs generated comparable heatmaps (Appendix A) with different ordering of the taxa, but with very similar clustering, successfully grouping the different serogroups. MPACT incorporates all the features available on SDT for nucleotide sequence identity percentage and also ML distance. In addition, MPACT can determine amino acid sequence identity and similarity, ML distance determined from an MSA using evolutionary models, and two metrics based on structural data: the TM-scores calculated from pairwise structural alignments, and 3Di-character pairwise sequence similarity (Appendix A). For all these metrics, MPACT can generate the respective heatmaps, UPGMA and NJ dendrograms, and frequency distribution plots, expanding the scope of analyses compared to SDT. In fact, MPACT allows for data partitioning utilizing user-defined upper and lower limits for each metric, providing significantly greater flexibility than SDT for taxa demarcation.

Dali is a program that performs a series of tasks, including structural comparisons of query 3D protein structures against a database of 3D structures. Among its several features, Dali allows one to run all-against-all pairwise structural alignments and generates heatmaps and dendrograms derived from average linkage clustering of the structural similarity matrix. The program offers a web server, limited to datasets of up to 64 sequences, or a standalone version that can be installed on a local server. We submitted a dataset of 64 3D structures, predicted by AlphaFold2 from RDRP sequences (Appendix A) of *Amalgaviridae* viruses, to an all-against-all analysis using MPACT and Dali. Appendix A shows the heatmaps obtained using the TM-score metric by MPACT and the corresponding results produced by Dali. Both programs generated similar heatmaps, featuring a large and easily identifiable cluster composed of members of the *Amalgavirus* genus, as well as several smaller clusters representing minor groups. In addition to Dali, MPACT can also provide comparative all-against-all analyses based on amino acid sequence identity and similarity, and ML distance determined by phylogenetic analyses. Also, 3Di-character sequence alignments can be obtained, increasing the scope of the analyses. Similarly to Dali, MPACT generates an NJ dendrogram, but can also provide dendrograms using a variety of different clustering methods, such as UPGMA and centroid.

## 4. Discussion

### 4.1. Using MPACT for Virus Comparison and Taxa Demarcation

In this work, we report the development of MPACT, the Multimetric Pairwise Comparison Tool, an integrated program that performs all-against-all pairwise comparisons using both primary biological sequences (nucleotide and amino acid) and 3D protein structures. Unlike the existing tools, MPACT is not restricted to a single method or metric; rather, it performs multiple analyses and provides a variety of outputs, including heatmaps, frequency distribution plots, and dendrograms obtained from various clustering methods. This broad set of analyses enables one to apply MPACT to a large gamut of viruses, allowing users to choose the most appropriate metric for determining relationships across viral groups. Notably, 3D structure similarity methods enable the comparison of distantly related viruses, effectively revealing relationships even across highly divergent viral groups, including the different families of the *Orthornavirae* kingdom (Appendix A) and the various subfamilies/groups of *Microviridae* phages (Figure 6). Additionally, MPACT implements the use of maximum likelihood (ML) distance, which, despite being a reductionist metric, relies upon phylogenetic reconstruction using evolutionary models. Our results confirmed that ML distance effectively unveils relationships across viruses with greater sensitivity than pairwise alignments, as demonstrated in the resulting heatmaps (Figure 4—*Amalgaviridae*; Figure 6—*Microviridae*) and corresponding dendrograms. Furthermore, ML distance-derived relationships can be used for data partitioning, with a high resemblance to taxa clustering from molecular phylogenetic methods. For example, the data partitioning of *Orthobunyavirus* nucleotide sequences (Appendix A) using ML distance closely matches the clades from ML phylogenetic reconstruction (Figure 7). Finally, the pairwise alignments of either nucleotide or amino acid sequences can be used as effective metrics to reveal relationships among closely related viruses. The capability to utilize multiple metrics based on pairwise alignment (identity and similarity percentage) and multiple sequence alignment (maximum likelihood distance), as well as structural data (TM-score and 3Di similarity percentage), renders the MPACT program significantly more flexible and broadly applicable than other available tools such as SDT and DALI.

### 4.2. Phenotypic and Genotypic Features for Viral Classification

Viral taxonomy was historically based on the Baltimore classification scheme [8], which initially comprised six viral groups, later expanded to seven, based on genome composition and viral replication strategies. In addition to the classical Baltimore classification, genotypic and phenotypic features can be used for taxonomic delineation.

Phenotypic characterization may more closely correlate with the biological features and have been used for taxa demarcation, especially to define viral species. Viral species have classically relied upon host specificity or range, cell and tissue tropism, mode of transmission, association to specific pathological/morbid entities, and ecological role of the viruses in a community, a particularly important feature for phages. Taxa demarcation based on phenotypical characteristics is subject to some limitations, including (1) subjectivity; (2) qualitative rather than quantitative features; (3) lack of sufficient knowledge of virus–host associations and specificities; (4) in vitro cultivation not available for many viruses; and (5) unknown environmental factor that may influence viruses.

In the case of genotypic classification, whole-genome sequences and/or their corresponding protein sequences can be employed in phylogenetic reconstruction using evolutionary models, and provide a high level of resolution and objectivity, providing a basis for phylogenetic classification. While this approach is being increasingly used, it is also subject to a series of potential limitations: (1) the typical high mutation rates of viruses, which may result in high divergence, leading to inconsistent alignments and trees; (2) no universal markers are available for viruses, that is, no single gene or protein is shared by all viruses, implying that specific markers must be used for particular viral groups; (3) horizontal gene transfer, recombination and segment reassortment events may complicate the analysis and lead to classification inconsistencies; (4) viruses may exhibit distinct evolutionary rates, as well as different molecular markers within the same taxa; (5) consistent evolutionary relationships require good taxa sampling, namely datasets that are representative of viral diversity; (6) the generation of reliable MSAs is increasingly difficult with large datasets and distantly related viruses; (7) establishing reliable thresholds of genetic distance can be tricky, especially for viruses presenting high sequence divergence or distinct evolutionary rates; (8) viruses may mimic their hosts through processes consistent with reticulate evolution; and (9) lack of correlation with phenotype, leading to classifications that show poor correlation with the biology of the viruses and their hosts. Some of these aspects have been used as key arguments in the ongoing debate on whether viruses should even be recognized as independent taxonomic units, defined by their unique organization, function, and evolution [73,74].

### 4.3. Single Metrics Versus Molecular Phylogeny

Phylogenetic reconstruction based on primary biological sequences is the golden standard for elucidating evolutionary relationships, as it is grounded in the established evolutionary models. However, the limitations discussed in the previous section present significant challenges to this approach. Achieving a comprehensive representation of viral diversity is highly desirable, yet processing large datasets for multiple sequence alignment (MSA) can be computationally demanding and introduce potential inaccuracies, creating an inherent conflict between data breadth and alignment accuracy. As the number of sequences in an MSA increases, the accumulation of indel events tends to reduce pairwise sequence identity/similarity scores [22] and the overall accuracy of the alignment.

In contrast, methods based on pairwise all-against-all comparisons, while affected by dataset size in terms of computational demand, are not disturbed by the size of input data. For example, SDT implements an all-against-all alignment of nucleotide sequences to determine identity percentage for all sequence pairs, a method proven valuable for the taxa demarcation of closely related viruses. Nevertheless, due to the typically high evolutionary rates of viruses, sequence identity percentages become less informative for distantly related taxa. Consequently, the use of more sensitive metrics, such as amino acid similarity percentage, ML distance, and, particularly, 3D protein structure alignments, significantly enhances the applicability of pairwise comparisons across a broader range of evolutionary distances.

In this study, we utilized MPACT with diverse metrics, analyzing sequence and structural datasets from highly divergent *Orthornavirae* viruses, moderately divergent *Microviridae* phages, and relatively closely related viruses of the genus *Orthobunyavirus*. As we presented and discussed in the preceding sections, distinct viral groups possess inherent specificities, thereby requiring particular approaches and criteria for the delineation of taxonomic boundaries. By using diverse methodologies, we observed that no single metric provides effective taxa demarcation and data partitioning that are perfectly congruent with classical molecular phylogeny. This is especially true when highly diverse viral groups are analyzed. In this case, phylogenetic analyses based on evolutionary models may provide significantly more information than any metrics discretized to single values.

Classifications based on reductionist metrics, regardless of their nature, may deviate from the true evolutionary relationships of the respective organisms. Consequently, any cutoff value for taxa demarcation is arbitrary and limited, potentially resulting in clusters that do not necessarily reflect evolutionary clades. Therefore, while specific lower and upper range limits for taxa demarcation may be effective for a particular viral group, they should not be indiscriminately extrapolated to diverse viral groups, as evolutionary rates vary among them, and consequently, so do the degrees of divergence across different metrics. We demonstrated that boundary definition becomes increasingly complex for more divergent viral groups, due to varying evolutionary rates across distinct lineages, leading to internal clade members potentially becoming more distant from each other than from external members.

Consequently, reliable demarcation is feasible for closely related viral genomes, such as those within the genus *Orthobunyavirus*, but becomes challenging for more distant groups, as exemplified by *Orthornavirae* (Appendix A) and *Microviridae* (Appendix A). In light of the debate on whether viruses should even be considered bona fide taxonomic units [74], species demarcation is still a controversial issue and perhaps remains a question that may prove objectively irresolvable. Considering that phylogenetic analysis does not necessarily cover the intricate relationship between viruses and their hosts, we agree with the principle that alternative categorizations based on infectivity and virulence may be more convenient for disease prevention and treatment, even though they do not reflect evolutionary relationships [1].

Although viral taxa demarcation may not perfectly reflect true evolutionary relationships, it does offer significant practical applications and benefits that should not be underestimated. It facilitates standardized viral nomenclature, data sharing, viral database management, and other crucial aspects. Rather than providing rigid foundations, taxa demarcation, based on selected metrics and criteria, should be viewed as a valuable framework for organizing and understanding the viral world, evolving dynamically as new information emerges. In this context, we believe that MPACT serves as a valuable tool for assisting the scientific community in delineating viral taxa across a spectrum of criteria. By providing heatmaps and distance trees derived from sequence and 3D protein structure data, MPACT offers multiple lines of evidence to support taxa demarcation, even for distantly related viral groups where traditional upper and lower threshold values may be ineffective.

### 4.4. Protein Structure Information and MSAs

Structural similarity, due to its higher conservation compared to primary sequence similarity [26], can assist in identifying distant homologs, as well as improve the quality of MSAs. Two distinct approaches have been reported in the literature to increment MSAs by using structural information. FoldMason [75] performs progressive multiple structural alignment (MSTA) using the 3Di-character alphabet developed for Foldseek [32]. This alphabet, comprising 20 discrete characters representing the local 3D arrangement of protein structures, reduces complexity, enabling the alignment of hundreds of thousands of protein structures within reasonable computational times. In addition to increased processing speed, Foldmason generates multiple structural alignments (MSTAs) that can improve the accuracy of phylogenetic analyses for distantly related proteins within the twilight zone. As an alternative to the 3Di-character alphabet, Edgar [76] developed a novel “mega-alphabet” in which each residue in the protein backbone is represented by a unique character, resulting in over 85 billion distinct states. The Reseek program, described in this report, not only generates this alphabet from 3D protein structures but also performs protein homolog detection with higher sensitivity and improved speed compared to other available tools such as TM-align, DALI, and Foldseek. The same author, who developed MUSCLE5 [77], an accurate multiple sequence aligner, recently implemented the “Muscle-3D” feature in MUSCLE5 (https://github.com/rcedgar/muscle, accessed on 1 March 2025), enabling the use of “mega-alphabet” input sequences and the generation of multiple structure alignments [78]. Once these alignments are obtained, MUSCLE5 converts them to conventional amino acid sequence alignments, which can be used by standard phylogenetic reconstruction tools. We intend to incorporate the utilization of FoldMason and Reseek/Muscle-3D in the future implementations of the MPACT program, aiming at improving the taxonomic classification of evolutionarily distant viruses.

### 4.5. For Viral Detection and Classification Go Shorter

Beyond viral taxonomic studies, viral classification based on genomic and metagenomic sequencing data is also crucial, especially for molecular epidemiology and virus surveillance. Our previous studies have demonstrated that viral detection and discrimination from metagenomic data are more accurate when performed using short, informative protein regions, rather than full-length sequences [6,7,79]. We recently described TABAJARA [25], a tool that implements various methods to select sequence alignment blocks conserved across all the sequences of an MSA or, alternatively, to identify blocks that are specific signatures of viral taxa. The program objectively determines such informative blocks, generates profile HMMs, and validates these models against training sets to estimate their sensitivity and specificity. By using customized cutoff scores, these profile HMMs, specific to a variety of viral taxa, can be used to detect and taxonomically classify viral sequences from metagenomic datasets. We envision a combined methodology that merges MPACT and molecular phylogeny to delineate viral taxa groups. For taxa discrimination and the inclusion of new viruses into established taxa, the use of informative regions instead of entire sequences or structures makes more sense by using profile HMMs designed for informative regions of each taxon.

## Figures and Tables

**Figure 1 viruses-17-00642-f001:**
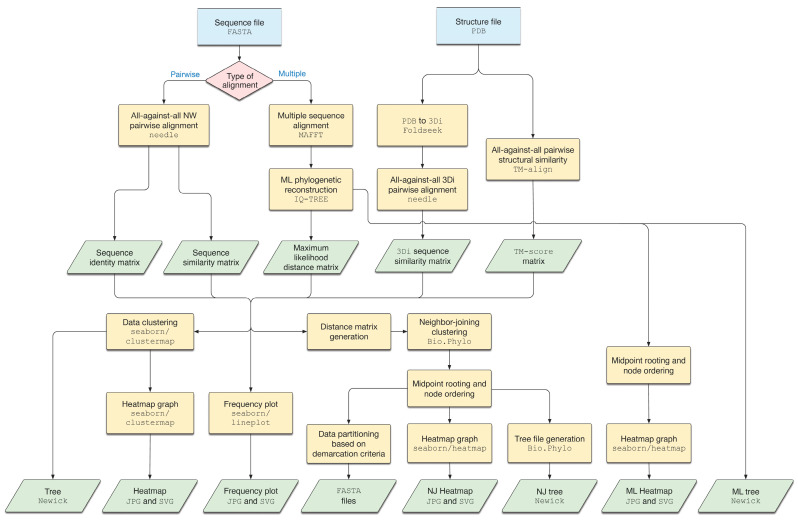
Workflow of the MPACT program. The program utilizes two types of input data: biological sequences in the FASTA format and 3D protein structures in the PDB format. The sequences are submitted to all-against-all pairwise alignments using the Needle program (EMBOSS package), and to a multiple sequence alignment (MSA) using the MAFFT program. The MSA is submitted to the IQ-TREE program for phylogenetic analysis, generating a maximum likelihood (ML) tree and a heatmap. Three-dimensional protein structures are subjected to all-against-all structural alignments using the TM-align program, and are also converted to 3Di-characters by Foldseek, and aligned with Needle. The alignment results are clustered and used to generate heatmap and frequency distribution plots. Also, distance matrices are produced and the data are clustered by the Neighbor-joining algorithm. The resulting tree is rooted at the midpoint and sorted in ascending order of the nodes. This order is used for data partitioning to generate groups of sequences selected within a range of user-defined values.

**Figure 2 viruses-17-00642-f002:**
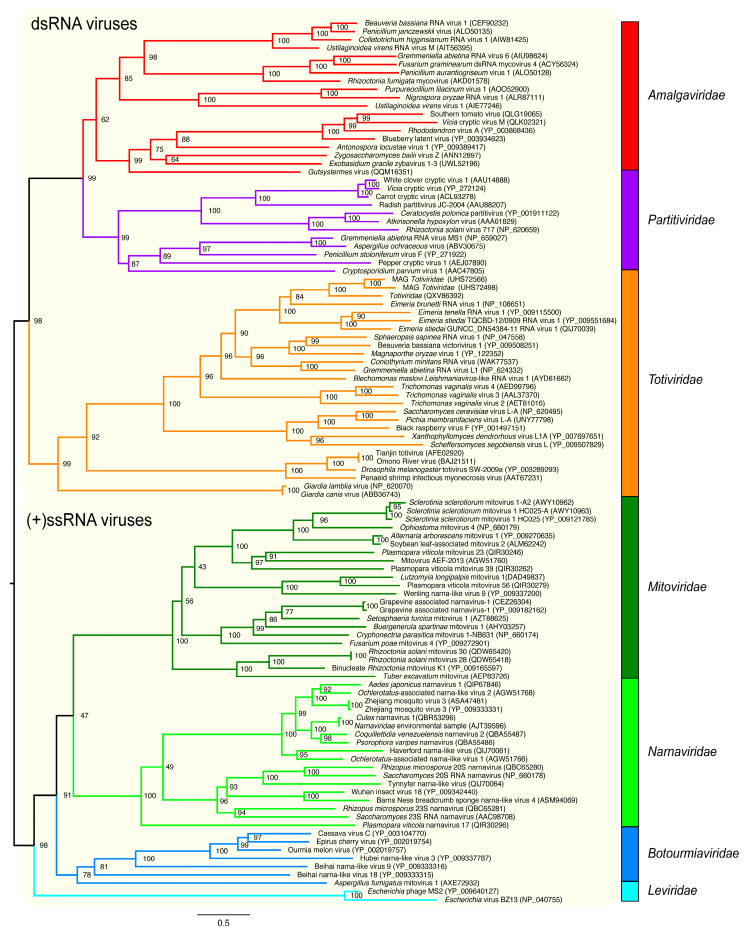
Phylogenetic reconstruction of RDRP amino acid sequences from representative taxa of different phyla of the *Orthornavirae* kingdom. The maximum likelihood tree was inferred using IQ-TREE 2 with the best-fit model VT + F + R7 on a multiple sequence alignment generated by MAFFT. The tree is rooted at the midpoint and the nodes are sorted in increasing order. The support values are shown at the nodes of the clades. The viral genome composition of the clades is depicted by the colored background. The colored vertical bars indicate the viral families of the respective clades.

**Figure 3 viruses-17-00642-f003:**
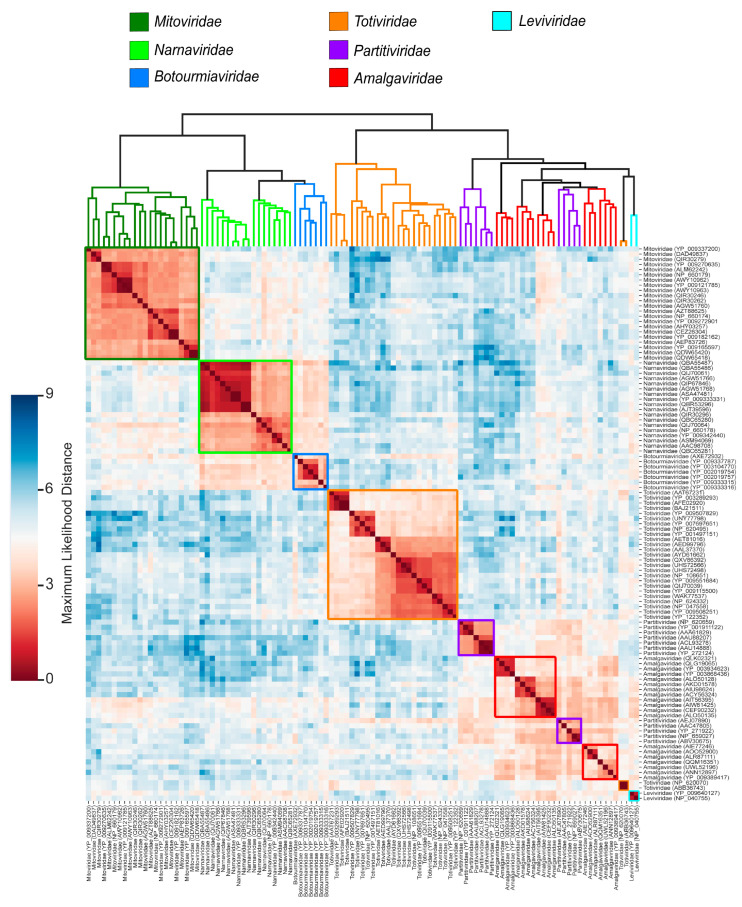
Heatmap of all-against-all pairwise comparisons of RDRP amino acid sequences from different phyla of the *Orthornavirae* kingdom. A multiple sequence alignment was performed using MAFFT and an ML distance matrix was obtained with IQ-TREE 2. The resulting ML distance values were clustered by the UPGMA method, and the upper dendrogram represents the clustered taxa. The data are displayed as a heatmap, with the upper right color scale representing the range of ML distance values. The colors of the clades in the tree and the colored squares on the heatmap represent different viral families.

**Figure 4 viruses-17-00642-f004:**
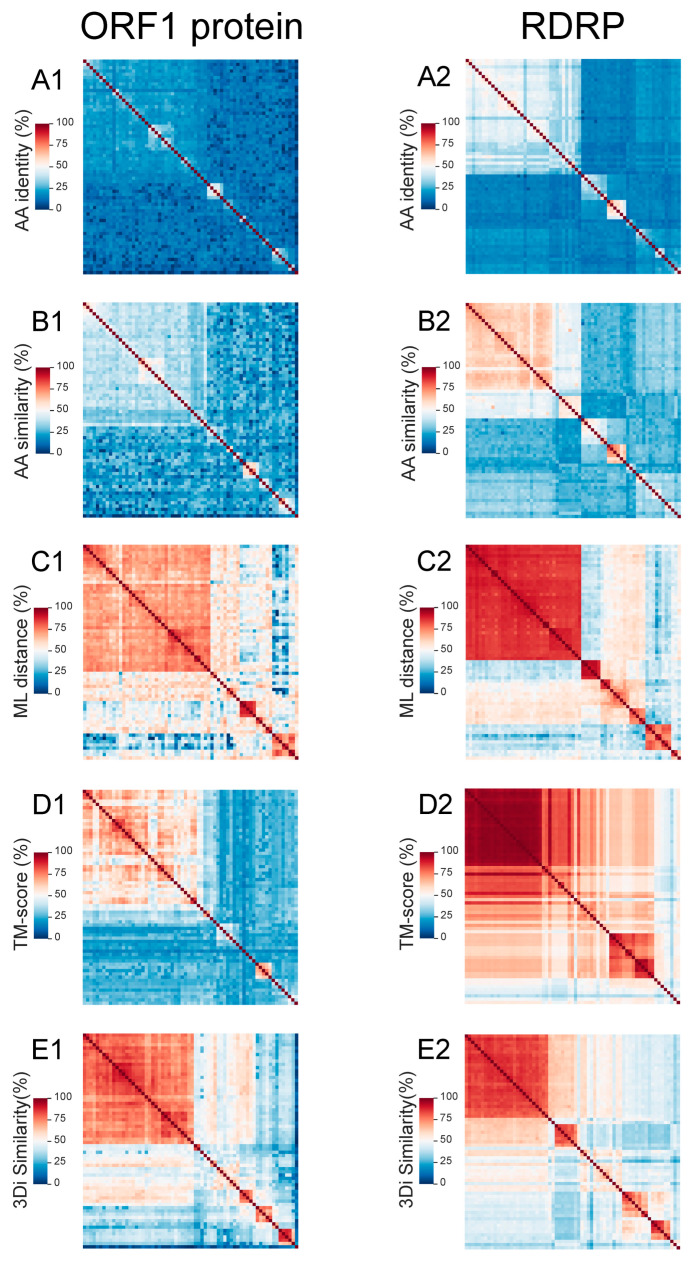
Heatmaps of all-against-all pairwise comparisons of the ORF1 protein (**A1**–**E1**) and RDRP (**A2**–**E2**) amino acid sequences of *Amalgaviridae* viruses derived from different metrics: identity (**A1**/**A2**) and similarity (**B1**/**B2**) percentages and maximum likelihood distance (**C1**/**C2**) of amino acid sequences, TM-scores of 3D structures (**D1**/**D2**), and 3Di-character sequence similarity (**E1**/**E2**).

**Figure 5 viruses-17-00642-f005:**
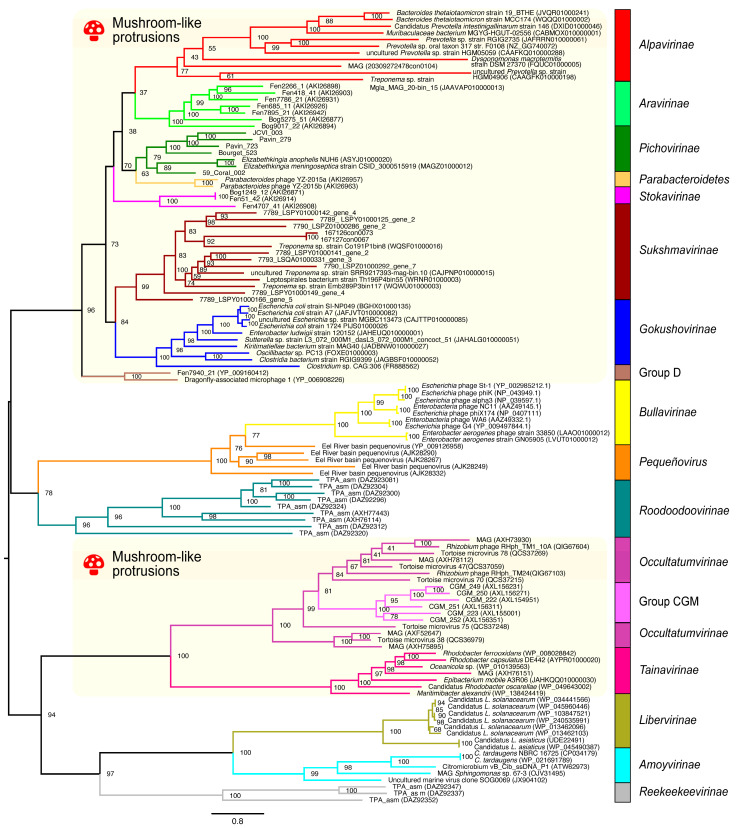
Phylogenetic reconstruction. Maximum likelihood tree inferred from amino acid sequences of the major capsid protein (VP1) derived from representative taxa of *Microviridae* viruses. The phylogenetic tree was obtained using IQ-TREE 2 with the best-fit model Q.pfam + F + R7 on a multiple sequence alignment generated by MAFFT. The tree is rooted at the midpoint and the nodes are sorted in increasing order. Support values are shown at the nodes of the clades. The colored background indicates clades whose viral members present mushroom-like protrusions as inferred from 3D structure predictions of the capsid proteins (VP1). The colored vertical bars indicate the viral subfamilies/groups of the respective clades. The names of the *Microviridae* groups/subfamilies used here are those described in the literature and are not officially recognized by the current ICTV classification.

**Figure 6 viruses-17-00642-f006:**
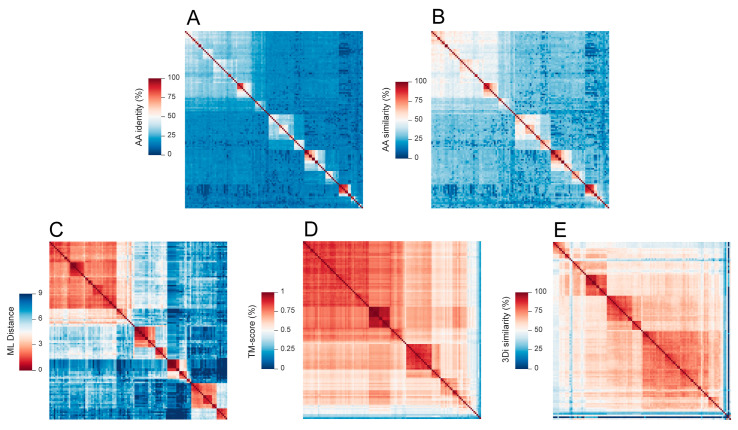
Heatmaps of all-against-all pairwise comparisons of VP1 amino acid sequences of *Microviridae* viruses derived from different metrics: identity (**A**) and similarity (**B**) percentages and maximum likelihood distance (**C**) of amino acid sequences, TM-scores of 3D structures (**D**), and 3Di-character sequence similarity (**E**).

**Figure 7 viruses-17-00642-f007:**
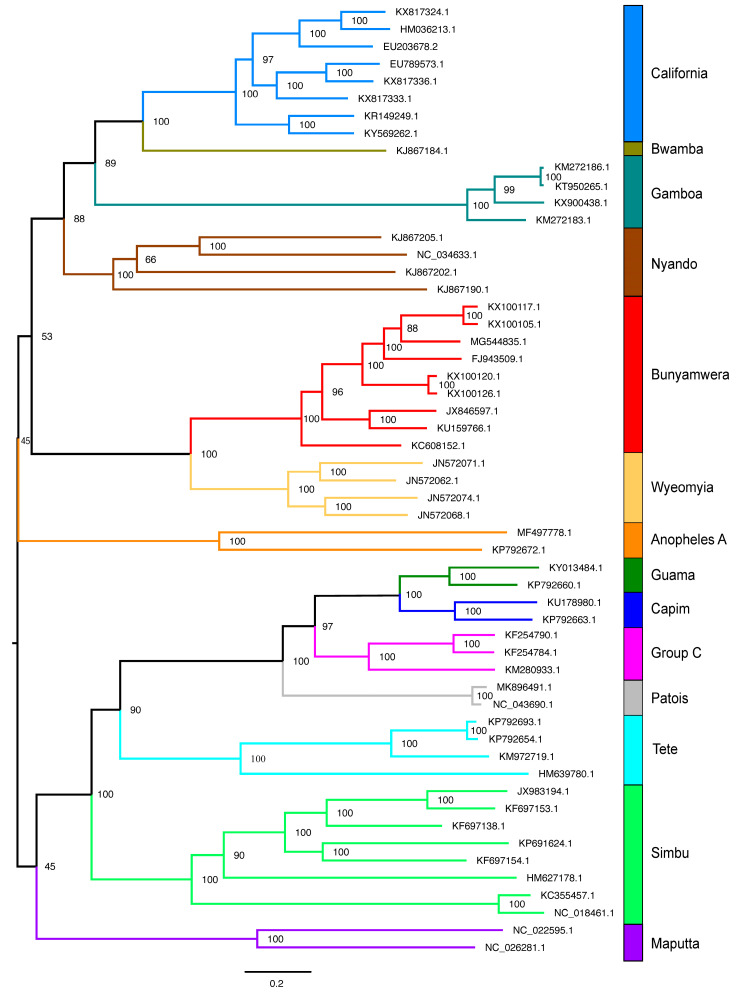
Phylogenetic reconstruction. Maximum likelihood tree inferred from nucleotide sequences of the L segment of viruses of the genus *Orthobunyavirus*. The phylogenetic tree was obtained using IQ-TREE with the best-fit model GTR + F + R7 on a multiple sequence alignment generated by MAFFT. The tree is rooted at the midpoint and the nodes are sorted in increasing order. Support values are shown at the nodes of the clades. The colored vertical bars indicate the antigenic groups of the respective clades.

**Figure 8 viruses-17-00642-f008:**
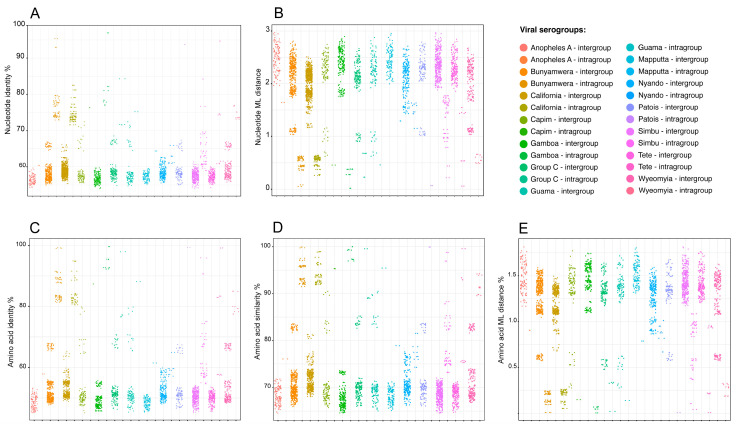
All-against-all pairwise nucleotide and amino acid comparisons of sequences derived from the large (L) segment of viruses of the genus *Orthobunyavirus*. Members of each viral group were defined by a maximum likelihood phylogenetic reconstruction using MAFFT and IQ-TREE, resulting in bona fide subsets. For each subset, sequences were compared all-against-all using pairwise identity percentage (**A**) and maximum likelihood distance (**B**) of nucleotide sequences, pairwise identity (**C**) and similarity (**D**) percentages, and maximum likelihood distance (**E**) of amino acid sequences. The dots of each column of the scatter plot depict the results obtained for intergroup and intragroup pairwise comparisons, respectively.

## Data Availability

MPACT (Multimetric Pairwise Comparison Tool) is an open-source program available for download in the GitHub repository (https://github.com/gruberlab/mpact, accessed on 2 April 2025), under the terms of the GNU General Public License version 3. The program is fully documented, and a tutorial is provided. Datasets used throughout this work, including nucleotide and protein sequences, and multiple sequence alignments are publicly available in the Appendix A.

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
