# Peer review of "Integrating Sequence- and Structure-Based Similarity Metrics for the Demarcation of Multiple Viral Taxonomic Levels"

_viruses, 2025, doi:10.3390/v17050642_

Round 1

Reviewer 1 Report

Comments and Suggestions for Authors The manuscript is about MPACT, a sequence comparison tool that integrates sequence and structure comparison for the taxonomic classification of viruses. This kind of integrated approach is new and relevant. MPACT combines the equivalent of SDT and Dali tools, which may be an improvement. As written, the paper is difficult to read because of some taxonomic confusions. Throughout the paper (mat met, results, discussion), different levels of taxonomy are mixed up to describe viruses and important names are missing. As a result, the paper lacks consistency and should be changed. The manuscript aims to describe a tool for virus taxonomists, so the current taxonomy should be better addressed. Three data sets are mentioned: -Orthornavirae is a kingdom name. It is used to describe only some Duplornaviricota (Totiviridae ), Pisuviricota ( Amalgaviridae and Partitiviridae) and Lenarviricota (Mitoviridae Botourmiaviridae, Narnaviridae and Leviviridae) without describing them. This is important because this classification is based on RdRp homologies and is therefore directly related to the authors' results in Figure 2 and should be taken into account when interpreting the results. It is also worth noting that this kingdom belongs to riboviria realm (RNA/RT virus) kingdom -Microviridae is a family name. To relate it to the kingdom level, it should be called Sangervirae. This kingdom belongs to the kingdom of Monodnaviria (ssDNA viruses). Here the authors use names of subfamilies that have been proposed in various publications but have not yet been approved by the International Committee on Taxonomy of Viruses (ICTV), which only recognizes Bullavirinae and Gokushovirinae. This decision by the authors is questionable. It would be better to use the official taxonomy first and then show that the results in Figure 5 are compatible with the proposed taxonomy from various publications. Non-official names should be marked with the addition "proposed" or some other indication that makes it clear that they are not official. Orthobunyavirus is genus name. It is confusing because it also belongs to orthornavirae kingdom. The latter is the name of the first dataset, so it is confusing in the text when the two are compared. It might be worth mentioning that it belongs to the Negaraviricota (ssRNA viruses), the kingdom of Riboviria. Material and method is a little short. The number of sequences used in the paper and their source are clearly indicated, but it is not clear where the "PDB" files come from. To my knowledge, there are very few experimental structures of the viruses used in the paper in PDB. I assume that the structures used by MPACT are structural predictions. If this is correct it should be indicated and an arrow from the sequence file to the structure file should be shown in Figure 1. the heatmaps in Figure 4 and 6 should be better labeled in the figure (identity, similarity,...) to facilitate interpretation.

Author Response

We thank the reviewer for carefully reading and analyzing our manuscript and for the valuable suggestions.

Comments 1: This kind of integrated approach is new and relevant. MPACT combines the equivalent of SDT and Dali tools, which may be an improvement.

Response 1:

We thank the reviewer for the positive feedback. MPACT indeed combines features from both SDT and DALI and introduces several additional capabilities. While SDT is limited to pairwise identity alignments of nucleotide sequences—and can be applied to protein sequences—it remains constrained to computing identity scores. As demonstrated in our manuscript, pairwise similarity often provides better sensitivity than identity, particularly for more distantly related sequences.

Compared to DALI, MPACT incorporates pairwise comparisons of 3Di characters, which under certain conditions can be more informative than full 3D structural alignments. We illustrate this with the Amalgaviridae ORF1 protein. Finally, MPACT also implements maximum likelihood distance calculations—a highly informative metric that is not available in either SDT or DALI.

Comments 2:

As written, the paper is difficult to read because of some taxonomic confusions. Throughout the paper (mat met, results, discussion), different levels of taxonomy are mixed up to describe viruses and important names are missing. As a result, the paper lacks consistency and should be changed. The manuscript aims to describe a tool for virus taxonomists, so the current taxonomy should be better addressed. Three data sets are mentioned: -Orthornavirae is a kingdom name. It is used to describe only some Duplornaviricota (Totiviridae ), Pisuviricota ( Amalgaviridae and Partitiviridae) and Lenarviricota (Mitoviridae Botourmiaviridae, Narnaviridae and Leviviridae) without describing them. This is important because this classification is based on RdRp homologies and is therefore directly related to the authors' results in Figure 2 and should be taken into account when interpreting the results. It is also worth noting that this kingdom belongs to riboviria realm (RNA/RT virus) kingdom.

Response 2:

All manuscript lines mentioned below refer to the originally submitted MS Word version of the manuscript.

In the item 3.2.1 of the Results section (lines 255-260) we stated: “we chose first to analyze a group of diverse RNA viruses of the Orthornavirae kingdom, comprising eukaryotic dsRNA viruses of the families Totiviridae and Amalgaviridae, presenting monopartite genomes, Partititiviridae, containing bipartite genomes, ssRNA positive-strand [(+)ssRNA] eukaryotic viruses of the families Mitoviridae, Narnaviridae and Botourmiaviridae, and two representatives of Leviridae, (+)ssRNA viruses that infect prokaryotes.”

We have shortly described the different families by mentioning the genome composition (dsRNA and ssRNA), the sense [(+)ssRNA] when appropriate, the monopartite and bipartite character of the genome, and the host range of the viruses (eukaryotic or prokaryotic). Nevertheless, we agree with Reviewer 1 that consistency and deeper descriptions are important, since the taxonomic groups are directly related to the results in Figure 2 and its interpretation. Following Reviewer 1’s recommendation, we increased the descriptions of all families, added the corresponding phylum names of these families, and included the information that the Orthornavirae kingdom belongs to the Riboviria realm.

Revised text:

“We chose first to analyze several RNA virus families belonging to some phyla of the kingdom Orthornavirae (Riboviria realm – RNA viruses and retroviruses). These included eukaryotic double-stranded RNA (dsRNA) viruses from the families Totiviridae (phylum Duplornaviricota), Amalgaviridae (phylum Pisuviricota) and Partitiviridae (phylum Pisuviricota). Both Totiviridae and Amalgaviridae have monopartite genomes, whereas Partitiviridae, commonly found in plants and fungi, typically possess bipartite genomes with two RNA segments. Totiviridae primarily infect fungi, protozoa, and invertebrates, and have genomes ranging from 4.6 to 7.0 kb that encode two open reading frames (ORFs): one for the capsid protein (CP) and another for the RNA-dependent RNA polymerase (RDRP). In Partitiviridae, the CP and RDRP are encoded on separate genome segments. While Totiviridae and Partitiviridae form non-enveloped icosahedral capsids approximately 30–40 nm in diameter, Amalgaviridae are believed to exist as non-encapsidated RNA–protein complexes within host cells. Amalgaviridae infect plants, fungi, and invertebrates, and their 3.4–3.5 kb genomes encode an ORF1 protein of unknown function and an RDRP. In addition, we included positive-strand single-stranded RNA [(+)ssRNA] eukaryotic viruses of the phylum Lenarviricota, comprising the families Mitoviridae, Narnaviridae, and Botourmiaviridae, as well as two representatives from the Leviridae family, which are (+)ssRNA viruses that infect prokaryotes. The three eukaryotic viral families have genomes ranging from 2 to 3.6 kb and each contains a single open reading frame (ORF) that encodes for the RNA-dependent RNA polymerase (RDRP). These viruses infect plants and fungi, and in the cases of Mitoviridae and Narnaviridae, they have also been found in invertebrates. Notably, no virions are produced by these viruses. Finally, we used representatives of the Leviviridae family as an outgroup. These viruses have an icosahedral capsid of 28 to 30 nm in diameter, contain a (+)ssRNA genome, and primarily infect prokaryotes, mainly bacteria.”

Comments 3:

Microviridae is a family name. To relate it to the kingdom level, it should be called Sangervirae. This kingdom belongs to the kingdom of Monodnaviria (ssDNA viruses). Here the authors use names of subfamilies that have been proposed in various publications but have not yet been approved by the International Committee on Taxonomy of Viruses (ICTV), which only recognizes Bullavirinae and Gokushovirinae. This decision by the authors is questionable. It would be better to use the official taxonomy first and then show that the results in Figure 5 are compatible with the proposed taxonomy from various publications. Non-official names should be marked with the addition "proposed" or some other indication that makes it clear that they are not official.

Response 3:

We followed the reviewer’s suggestion and added the taxonomic information to Microviridae (Monodnaviria - ssDNA viruses, Sangervirae kingdom). To clarify the nomenclature used for Microviridae viruses, it's important to note that the ICTV classification of this group has remained largely unchanged for many years, leaving numerous viral groups described over the past decade as 'unclassified'. Since 1978 (MSL #05), the Microviridae classification has only been updated once, in 2019 (MSL #35), with the establishment of a megataxonomic framework that filled all principal taxonomic ranks for ssDNA viruses. Furthermore, the 2024 release (MSL #40) only ratified the 1978 classification. Regarding the lower taxonomic groups of the Microviridae family, Gokushovirinae was ratified in 2009 (MSL #25) and Bullavirinae in 2015 (MSL #30). Thus, ICTV has not yet recognized the many groups described in the literature for this highly diverse family for more than a decade.

In our opinion, summarizing a set of 17 well-described viral groups with consistent phylogenetic relationships into only two officially recognized taxonomic groups would be uninformative for the reader. This decision is corroborated by numerous publications that have also presented phylogenetic trees depicting the various Microviridae groups/subfamilies using their literature-proposed names.

Nevertheless, we concur with Reviewer 1 that the official classification should be mentioned. Accordingly, we have revised the Materials and Methods section.

“A dataset composed of 119 sequences of the major capsid protein (VP1) from representatives of the different groups/subfamilies of the Microviridae family reported and proposed in the literature, not officially recognized by the current ICTV classification.”

Furthermore, we have added the following sentence to the legend of Figure 2: "The names of the Microviridae groups/subfamilies used here are those described in the literature and are not officially recognized by the current ICTV classification."

Comments 4:

Orthobunyavirus is genus name. It is confusing because it also belongs to orthornavirae kingdom. The latter is the name of the first dataset, so it is confusing in the text when the two are compared. It might be worth mentioning that it belongs to the Negaraviricota (ssRNA viruses), the kingdom of Riboviria.

Response 4:

Following the reviewer suggestion, we clarified this point by changing the titles of the sections that describe the different datasets:

From 2.1.1 RNA viruses of Orthornavirae

To 2.1.1 RNA virus families of the Orthornavirae kingdom (Riboviria realm – RNA viruses)

From 2.1.2 Microviridae

To 2.1.2 Microviridae family (Monodnaviria  realm - ssDNA viruses).

From 2.1.3 Orthobunyavirus

To 2.1.3 Orthobunyavirus genus (Negaraviricota kingdom – negative-strand ssRNA viruses, Riboviria realm)

Also, the descriptions of the datasets in the respective section also incorporate these taxonomic informations.

Comments 5:

Material and method is a little short. The number of sequences used in the paper and their source are clearly indicated, but it is not clear where the "PDB" files come from. To my knowledge, there are very few experimental structures of the viruses used in the paper in PDB. I assume that the structures used by MPACT are structural predictions.

Response 5:

We believe the Materials and Methods section should describe the methodology with sufficient detail to allow complete and exact reproduction of the experiments. To this end, we provide not only the complete sequence datasets, but all accession codes in the supplementary material. The GitHub repository where the MPACT code is available for download includes all sequence and 3D structure datasets and a detailed tutorial for users to exactly reproduce all the experiments performed and described in this work. Unless otherwise specified by Reviewer 1, we believe this section contains the necessary information for the reader.

Specifically, regarding the PDB files, section 2.3 specifies that the protein sequences of the datasets have been used to predict their respective 3D structures using the AlphaFold2 platform (lines 199-204). PDB is the format of the files generated by the AlphaFold2 platform, which was used in this study to predict the 3D structures of the proteins. Therefore, we believe that the origin of the PDB files used in this work has been described.

The reviewer's assumption that the structures used by MPACT are structural predictions is correct. This aspect is described in the text of section 3.1 (lines 226-228 and 234-239).

Comments 6:

If this is correct, it should be indicated, and an arrow from the sequence file to the structure file should be shown in Figure 1.

Response 6:

Figure 1 depicts the workflow of the program MPACT and is a graphical description of all processing steps executed by the program. MPACT does not perform the 3D structure prediction itself, the same way as it does not convert raw data from the DNA sequencer into FASTA sequences. For this reason, this task is not included in the workflow. Rather, the workflow shows that the FASTA sequence files and the PDB structure files represent two independent types of data (labeled in blue boxes in the diagram and detailed in the legend) that must be supplied as input.

Comments 7:

The heatmaps in Figure 4 and 6 should be better labeled in the figure (identity, similarity,...) to facilitate interpretation.

Response 7:

In both figures, the metrics are listed with their corresponding letter in the respective legends. Furthermore, all color scale legends also include this information (identity, similarity,...). We believe adding this description to each heatmap in an additional place would be redundant.

Reviewer 2 Report

Comments and Suggestions for Authors

dos Santos et al., present a new bioinformatics tool, MPACT that utilises a number of metrics to enable investigation the demarcation of viral taxa. The manuscript is very well written and comprehensive, providing a substantial amount of data and supplementary material. Other tools are appropriately described in the introduction and a more detailed comparison to SDT and Dali is presented in the manuscript. The discussion of the results is, in my opinion, well considered and measured.

I was able to install MPACT and run through the tutorial without any issues on a Linux server.  

One minor comment would be to  be specific where comparisons are described for nucleotide and amino acid sequences. For example, at lines 442, 490 and 498 the authors could qualify the genes used within the comparison (e.g. the large segment encoding the RNA-directed RNAP for Orthobunyaviruses). Were any other genes considered for analysis in the Microviridae and Orthobunyavirus as it might have been of interest to compare the results obtained for these?

Aside from the minor comments above, I am of the opinion that manuscript helps to further the field and is appropriate for publication.

Author Response

COMMENTS 1: dos Santos et al., present a new bioinformatics tool, MPACT that utilises a number of metrics to enable investigation the demarcation of viral taxa. The manuscript is very well written and comprehensive, providing a substantial amount of data and supplementary material. Other tools are appropriately described in the introduction and a more detailed comparison to SDT and Dali is presented in the manuscript. The discussion of the results is, in my opinion, well considered and measured.

 RESPONSE 2:

We thank the reviewer for the positive comments.

COMMENTS 2:

I was able to install MPACT and run through the tutorial without any issues on a Linux server.  

RESPONSE 2:

We thank the reviewer for testing the program and reporting that it functioned correctly.

COMMENTS 3:

One minor comment would be to be specific where comparisons are described for nucleotide and amino acid sequences. For example, at lines 442, 490 and 498 the authors could qualify the genes used within the comparison (e.g. the large segment encoding the RNA-directed RNAP for Orthobunyaviruses).

RESPONSE 3:

We thank the reviewer for the suggestion, and we have added this information in multiple instances along the text and in the legends of the figures and tables of the Supplementary Material.

COMMENTS 4:

Were any other genes considered for analysis in the Microviridae and Orthobunyavirus as it might have been of interest to compare the results obtained for these?

RESPONSE 4:

To validate the program, we employed multiple viral datasets encompassing diverse taxonomic groups and levels, including nucleotide and protein sequences: Orthornavirae RDRP; Microviridae VP1; Amalgaviridae ORF1 protein and RDRP; and Orthobunyavirus RDRP and L segment. Given that we used phylogenetic reconstructions as our reference standard to guide our analyses, we primarily selected classical phylogenetic markers such as RDRP and VP1, with the exception of the ORF1 protein in Amalgaviridae. We chose to test this protein specifically because it was long assumed to be a capsid protein, despite the absence of observed viral particles in this family. The protein's high alpha-helix domain content, coupled with low confidence in its predicted 3D structure, presented a compelling case study.

In our opinion, the current body of evidence presented in this work is already very extensive, and the inclusion of additional genes would make the manuscript excessively long without necessarily enhancing the demonstration of MPACT's applicability. Nevertheless, we do agree with the Reviewer that these analyses would be highly valuable within the context of more in-depth investigations of the diverse viral groups examined herein, a task that we intend to do in future work.

Round 2

Reviewer 1 Report

Comments and Suggestions for Authors

Thank you for your answers